# Clinical outcomes and mortality in old and very old patients undergoing cardiac resynchronization therapy

**Luiz Eduardo Montenegro Camanho**[1,2]ᵒ *, **Eduardo Benchimol Saad**[2]ᵒ, **Charles Slater**[2‡], **Luiz Antonio Oliveira Inacio Junior**[2‡], **Gustavo Vignoli**[2‡], **Lucas Carvalho Dias**[2‡], **Pedro Pimenta de Mello Spineti**[1‡], **Ricardo Mourilhe-Rocha**[1,2]ᵒ

1 Serviço e Disciplina de Cardiologia, Universidade do Estado do Rio de Janeiro, Rio de Janeiro, Brasil,
2 Serviço de Cardiologia e Arritmia Invasiva, Hospital Pró-Cardíaco, Rio de Janeiro, Brasil

ᵒ These authors contributed equally to this work.
‡ These authors also contributed equally to this work.
* lecamanho@globo.com

## Abstract

### Aim

Cardiac resynchronization therapy (CRT) is a valid therapeutic option for patients with heart failure (HF). However, the elderly population was not well represented in the guidelines. The primary end point was to evaluate the impact of advanced age on clinical response and cardiovascular and total mortality of patients undergoing CRT. The secondary end point was to assess the rate of acute complications related to the procedure.

### Methods and results

A total of 249 consecutive patients with HF and optimized treatment, QRS $\geq$ 120 ms, ejection fraction (EF) $\leq$ 35% and functional class (FC) III/ IV (NYHA) underwent CRT and divided into 3 groups: Group I—< 65 years—88/ 249 (35%); Group II– 65 to 75 years (old)– 72/ 249 (29%); Group III—$\geq$ 75 years (very old)– 89/ 249 (36%). The improvement in FC and increase in EF (>10%) and/ or decrease in the left ventricular end systolic diameter (LVESD) >15% were the criteria of responsiveness. The favorable clinical response (p = ns) and cardiovascular mortality (p = 0.737) was similar in the 3 groups. In the group of very old patients, a significant increase in total mortality was observed (p = 0.03). The rate of acute complications related to the procedure did not differ between the groups (p = ns).

### Conclusion

The response to CRT and cardiovascular mortality were not affected by the advanced age and should not be an exclusion factor of this therapy. The procedure has been shown to be safe in elderly patients due to low rate of acute complications.

**Data Availability Statement:** All relevant data are within the manuscript (Results section).

**Funding:** The authors received no specific funding for this work.

**Competing interests:** The authors have declared that no competing interests exist.

## Introduction

Advanced HF is a condition with a progressively increasing prevalence worldwide. It is estimated that the prevalence of this condition will increase by 46% between 2010 and 2030, probably related to increased life expectancy [1,2]. Advanced age is an important risk factor for HF, which is one of the main causes of hospital admission in the elderly. Approximately 2% of all adults in developed countries have HF. Most of them are aged > 70 years, and it is estimated that half of this population present HF with ejection fraction (EF) <50% [3]. The 1-year mortality of patients hospitalized for HF is around 20% and, in those aged >75 years, it reaches 40% despite optimized pharmacological therapy [4,5]. Cardiac resynchronization therapy has demonstrated a beneficial impact on the symptomatology and quality of life, as well as in reducing complications and mortality in patients with left ventricular systolic dysfunction and advanced functional class (FC) [6–8]. However, the elderly population is not included in the majority of current clinical trials and guidelines, in which the mean age is generally <70 years [6–9]. In the guidelines, there is no upper limit for the age of patients eligible to receive an implantable CRT device or an implantable cardioverter defibrillator (ICD); however, placement of such devices are recommended to be avoided in elderly and frail patients, and in those with a life expectancy of <1 year. The reasons of this recommendation include the frequently associated comorbidities, risk of complications related to an invasive procedure, and difficulty of access to specialized services in patients with HF [10].

## Methods

### Patient selection

The study population originally included 249 consecutive patients with advanced HF and successfully implanted with a biventricular pacing device (CRT-P) with or without an ICD (CRT-D). Enrollment started in March 2006 and ended in June 2015 and data were recruited from June to December 2018. The study was conducted in early 2019. The inclusion criteria were advanced HF with optimized pharmacological therapy, EF ≤35%, QRS ≥120 ms, and New York Heart Association (NYHA) FC III/IV. The exclusion criteria were severe clinical disease with a life expectancy of <1 year and symptomatic bradyarrhythmia with a narrow QRS complex. Optimized pharmacological therapy included, whenever possible, the following drugs: beta-blockers, angiotensin-converting enzyme inhibitors (ACEIs) or angiotensin receptor blockers (ARBs), aldosterone antagonists, and hydralazine and nitrate (for those with contraindication to ACEIs or ARBs). The use of diuretics or digoxin was dependent on the clinical profile of each patient.

The patients were divided into 3 groups: group I–age < 65 years, 88/249 (35%); group II–age 65–75 years (old), 72/249 (29%); and group III–age ≥ 75 years (very old), 89/249 (36%). Clinical evaluation and transthoracic echocardiography were performed 3 months after the implantation to assess the responsiveness of the patients. At every 6 months follow-up visit, device interrogation was performed, which included evaluation of the system and lead information (thresholds, amplitude, and impedance). CRT-D was the prosthesis used in 75% of the patients, comprising 69/88 patients (78%) from group I, 55/72 patients (76%) from group II, and 65/89 patients (73%) from group III.

The CRT response criteria were the improvement of FC and increase in EF (>10%) and/or decrease in left ventricular end-systolic diameter (LVESD) >15%. The mean follow-up time was 38.9 months.

The primary endpoint of the study was to evaluate the clinical response and cardiovascular and total mortality in the 3 groups of patients who underwent CRT. Cardiovascular mortality

was characterized as death secondary to ischemic or hemorrhagic stroke, acute myocardial infarction, sudden death, or pump failure (cardiogenic shock). The secondary endpoint of the study was to define the rate of acute and procedural complications in the 3 groups. The occurrence of pneumothorax, cardiac tamponade, or lead dislodgment was defined as an acute complication.

The present study was conducted in accordance with the Declaration of Helsinki, and all patients signed the informed consent form before undergoing the procedure. The authors did not have access to information that could identify individual participants during or after data collection.

## Eletrocardiography

A 12-lead electrocardiography with N pattern (10 mm/mV) and a velocity of 25 mm/s was performed before the procedure and at each follow-up visit. The presence of sinus rhythm or atrial fibrillation was evaluated. In the group with atrial fibrillation, it was decided whether or not ablation of the atrioventricular (AV) node was needed to maintain biventricular stimulation. The duration of QRS in milliseconds and its morphology were also evaluated to characterize the presence of left bundle branch block, right bundle branch block, or other interventricular conduction disturbances.

## Echocardiography

Echocardiographic evaluations were performed before and at 3, 6, and 12 months after device implantation. In patients with prolonged follow-up, at least 1 annual echocardiographic examination was performed. The examination was performed with the patient at rest in the left lateral decubitus position. Images of the 2-dimensional echocardiogram with continuous and color Doppler were obtained.

The EF of the left ventricle was estimated using the Simpson method. The final systolic and diastolic diameters of the left ventricle were measured in millimeters. The degree of mitral regurgitation was considered severe when the following criteria were met: regurgitant area $>40\%$ of the left atrium area ($cm^2$), regurgitant volume $\geq 60$ mL/beat, regurgitant orifice area $\geq 0.4$ $cm^2$, and regurgitant fraction $\geq 50\%$.

## Cardiac resynchronization therapy device implantation

All patients were implanted with a biventricular pacemaker (CRT-P or CRT-D from Biotronik [Berlin, Germany], Boston Scientific [Marlborough, MA, USA], Medtronic [Minneapolis, MN, USA], and St Jude Medical [Saint Paul, MN, USA]). Venous access was obtained through puncture and/or dissection of the axillary vein, in a total of 3 accesses. In the last 36 cases, the axillary vein puncture was guided with ultrasound.

The right atrium and right ventricle were stimulated by positioning standard leads in the right atrial appendage and right ventricular septum, respectively.

A combined device (CRT plus internal defibrillator) was implanted in 189 patients (75%). Device implantation was successful in all cases, with a low occurrence of major complications.

The standard settings included an AV delay of 120 ms (paced) and 100 ms (sensed), DDD or DDDR mode (VVI or VVIR if permanent atrial fibrillation was present), a lower pacing rate of 60 bpm, and an upper pacing rate of 130 bpm. The AV interval and VV intervals were adjusted for optimal diastolic filling and left ventricular outflow assessment with Doppler echocardiography.

## Statistical analysis

The collected data were descriptively analyzed for each age group and for the total sample. The means, medians, standard deviations, and interquartile ranges were calculated for numerical variables. For categorical variables, the counts and percentages of the responses were calculated.

To verify the association of some clinical markers with the analyzed groups, we employed techniques of inferential statistics through hypothesis tests. We used the chi-square test, Fisher's exact test, t-test, Mann-Whitney test, 1-factor analysis of variance, and Kruskal-Wallis test according to the necessary assumptions.

To meet the main objective of the study, we also verified whether there were multiple factors associated with the CRT response of the patients. For this analysis, logistic regression was performed with a stepwise methodology for the selection of variables.

The confidence level of the inferential analyses was 95%, and the statistical software used was R version 3.5.1. Values of $p < 0.05$ were considered statistically significant.

## Results

### Study population and baseline characteristics

Of the overall population (249 consecutive patients), 170 (68.5%) were men. The mean age of the patients was 69.1 years. HF was mainly due to ischemic cardiomyopathy (130 patients, 52.2%) with a mean EF of 29% (24–32%) in the whole population. A total of 230 patients (92.4%) had NYHA FC III at the time of implantation. Left bundle branch block was found in 228 patients (95.5%). The baseline characteristics of the study population, also divided by subgroups, are summarized in Table 1. The study subgroups were composed of 88 (35%) non-elderly patients (age <65 years), 72 (29%) old patients (age between 65 and 75 years), and 89 (36%) very old patients (age >75 years). The CRT responsiveness and cardiovascular and total mortality were analyzed, in addition to evaluating the rate of acute complications related to the procedure.

### Primary endpoint: cardiac resynchronization therapy response and mortality

The response to CRT was not directly affected by advanced age. When the response based on FC improvement was analyzed, the observed rate in the 3 groups was 81.8%, 94.4%, and 91%, respectively (mean, 89%). However, when using the criterion of FC improvement and at least 1 echocardiographic criterion, the observed response rate was 52.3%, 56.9%, and 68.5%, respectively (mean, 59.2%). The cardiovascular mortality rate was also not affected by advanced age (S1 Fig), with no statistical difference among the 3 groups ($p = 0.737$). Only the total mortality (S2 Fig) was higher in the group of patients aged >75 years (very old; $p = 0.03$). In this group (>75 years), ischemic etiology, defined as previous myocardial infarction, PTCA, myocardial revascularization or severe coronary obstructive disease, was observed in 53.9%. No clinical differences and outcomes were observed between ischemic and non-ischemic very old patients.

CRT-D was the prosthesis used in 75% of the patients approximately. When survival curves were analyzed according to the type of prosthesis, it was observed that CRT-p or CRT-d were not predictors of cardiovascular or total mortality, with similar survival rates.

The LBBB occurred in 93%, 94% and 98%, respectively, in the three groups. In 12/ 249 pt (4.8%) was observed right bundle branch block (RBBB), 6 in group I, 2 in group II and 4 in

**Table 1. General characteristics of the whole population.**

| | Grupo I (<65 y) | Grupo II (65–75 y) | Grupo III (> 75 y) | p |
|---|---|---|---|---|
| Age (years) | 54.9 ± 9.4 | 70.5 ± 3.1 | 82.1 ± 4.3 | <0.001 |
| Male sex (n,%) | 59 (67.8) | 49 (68.1) | 62 (69.7) | 0.96 |
| CKD (n,%) | 2 (2.5) | 6 (8.8) | 15 (18.1) | 0.004 |
| Diabetes melittus (n,%) | 27 (31.8) | 32 (44.4) | 35 (40.2) | 0.245 |
| Atrial fibrillation (n,%) | 24 (27.3) | 12 (16.7) | 27 (30.3) | 0.122 |
| COPD (n,%) | 3 (3.5) | 4 (5.6) | 7 (8.0) | 0.443 |
| Hypertension (n,%) | 54 (63.5) | 64 (88.9) | 74 (84.1) | < 0.001 |
| Syncope (n,%) | 5 (5.9) | 7 (9.9) | 4 (4.5) | 0.385 |
| Functional class III (n,%) | 80 (90.9) | 66 (91.7) | 84 (94.4) | 0.661 |
| Ischemic etiology (n,%) | 35 (39.8) | 47 (65.3) | 48 (53.9) | 0.005 |
| LBBB (n,%) | 82 (93%) | 68 (94%) | 88 (98%) | 0.024 |
| Ejection fraction (%) | 27 (21.7–31.2) | 28 (24–31) | 30 (26–32) | 0.014 |
| LVESD (mm) | 59 (52.2–63) | 55 (49–61) | 52 (49–59) | < 0.001 |
| LVEDD (mm) | 73.35 ± 8.12 | 70.85 ± 5.97 | 67.44 ± 7.54 | < 0.001 |
| QRS width (ms) | 160 (152–180) | 160 (160–180) | 160 (160–180) | 0.323 |
| Severe MR (n,%) | 19 (21.8) | 7 (10.3) | 6 (7.1) | 0.013 |

Continuous variables are expressed as mean, standard deviation and median. Categorical variables are expressed as total number (%). CKD, chronic kidney disease; COPD, chronic obstructive pulmonary disease; LBBB, left bundle branch block; LVESD, left ventricular end systolic diameter; LVEDD, left ventricular end diastolic diameter; Severe MR, severe mitral regurgitation.

group III. The QRS width average was 167 ms. Except one patient in group I, all RBBB pt were responders to CRT.

## Secondary endpoint: acute complications related to the procedure

Acute and procedural complications were those that occurred within the first 24 h, including pneumothorax, cardiac tamponade, and lead dislodgment (Table 2). No statistical differences were observed in any of the variables among the 3 groups (p = ns), demonstrating the safety of the procedure even in very old patients.

## Discussion

In the present study, patients who underwent CRT-P or CRT-D implantation were divided into 3 groups according to age: <65 years, between 65 and 75 years, and >75 years, therefore including a population of old and very old patients. For the primary endpoint, the CRT response (based on FC improvement and at least 1 echocardiographic criterion) and cardio-vascular mortality were not affected by advanced age. There was a statistically significant increase in total mortality in the very old patient group, which is an expected outcome in an age group with increased prevalence of other morbid clinical conditions.

**Table 2. Acute complications related to the procedure.**

| | Group I | Group II | Group III |
|---|---|---|---|
| Pneumotorax | 3 (3.4%) | 2 (2.7%) | 2 (2.2%) |
| Cardiac tamponade | 1 (1.1%) | 0 | 0 |
| Lead dislogdment | 3 (3.4%) | 2 (2.7%) | 3 (3.3%) |

Advanced HF is a serious public health problem, especially in developed countries and with a progressively older population. In persons >65 years old, HF is the cause of hospital admission in 20% of the cases and patients >80 years old are 20 times more likely to be admitted for HF than patients aged 36–64 years [11]. The mortality from HF in this age range is also higher, independent of optimized pharmacological therapy [12].

The vast majority of trials and medical guidelines for CRT do not include the elderly population. The mean age of patients enrolled in the CARE-HF, MIRACLE, MIRACLE-ICD, COMPANION, MADIT-CRT, and RAFT studies ranged from 63.9 to 67 years [7,8,13–16]. In the CARE-HF study [7], only 6.1% of the patients were >80 years old. Previous small studies in CRT recipients ≥80 and <80 years old demonstrated similar clinical outcomes regardless of age range [17,18].

Bleeker *et al* was the first to describe the results of CRT in 170 consecutive patients who were divided into 2 groups according to age: ≥70 and <70 years. The clinical and echocardiographic improvements at 6 months and the survival at 2 years were similar between the 2 groups [19]. However, the pharmacological therapy and rate of hospitalization due to HF were not described in this study. Another smaller study compared 36 patients aged ≥65 years and 51 patients aged <65 years undergoing CRT. The reverse remodeling rate was similar between the 2 groups at the end of 6 months [20].

In 2013, Verbrugge *et al* described 220 consecutive patients who underwent CRT and were divided into 3 groups according to age: <70 years, 70–79 years, and ≥80 years. Although it was a retrospective analysis and there was a significantly shorter QRS duration in the population of patients aged <70 years, the reverse remodeling rate and improvement of functional capacity were similar among the 3 groups despite the advanced age of the population. Similarly, the total mortality and hospitalization rates were not affected by age [21].

Höke *et al* evaluated the response to CRT, adverse events, and long-term results of CRT in elderly patients. A total of 798 patients were divided into 2 groups according to age: ≥75 years (208 patients) and <75 years (590 patients). The efficacy of CRT and the incidence of prosthesis-related complications were similar between the 2 groups. However, after 4 years of follow-up, the survival of elderly patients was lower, although owing to non-cardiac causes. Diabetes mellitus, chronic kidney disease, and worse performance in the 6-min walk test were independent factors of total mortality in the population aged >75 years [22]. Similarly, Kowalik *et al* recently described the clinical factors associated with the long-term survival of 223 patients undergoing CRT and aged > 70 years. In the population with advanced age (>70 years), the only independent factor of worse prognosis was renal failure, with no negative impact according to the age group [23].

Our results are in agreement with the findings in the literature, in which an increase in total and non-cardiovascular mortality was observed in the population of very old patients (age >75 years). Similarly, the response to CRT was also not affected by advanced age (old and very old) in our study.

Although CRT has been a well-established method in medicine for >20 years, there is still no consensus about the definition of "response" and "no response" in relation to this therapy. Several ways of measuring treatment response are commonly used in clinical trials, including assessment of functional criteria (FC and quality of life), hard outcomes (total mortality and hospitalization due to HF), echocardiographic parameters of reverse remodeling, and clinical composite outcomes. The non-response rate tends to be lower when using only the functional evaluation criterion (functional class). Although comparative studies do not demonstrate a direct correlation between clinical response and echocardiographic response, when a reverse remodeling criterion is associated with CRT assessment, the non-response rate was described to range from 35% to 40% [24,25]. Our results reflect this situation and the need for a more

definitive classification of the problem, as when using only the FC, the response rate to CRT in the 3 groups was 82.8%, 94.9%, and 91%, respectively. When an echocardiographic reverse remodeling criterion was included, the response rate was 52.3%, 56.9%, and 68.5%, respectively.

Although there are >3500 scientific publications on ICD/CRT device implantation in patients aged >65 years, there are around 20 publications about the safety of the procedure in patients aged >75 years [26,27]. Of these, only 5 studies evaluated safety and complications directly related to the procedure (in 4 studies, the authors used the limit of 80 years to define advanced age; in the other study, the age limit was 75 years). In these studies, there was no statistically significant difference between general adverse events and procedure-related events among young and elderly patients [10,18,21,22,28]. Only the study by Olechowski *et al*, in which 439 patients (aged ≥80 and <80 years) who underwent CRT were analyzed, concluded that the procedure is feasible and safe in very old patients with multiple comorbidities. However, there was a significantly higher occurrence of pneumothorax in patients ≥80 years [10].

The use of ultrasound-guided axillary or cephalic vein access has been increasingly established as a safe and effective technique related to lower rates of acute complications [29,30].

Our results are important in describing acute complications related to CRT, considering the scarcity of data in the current literature. The occurrence of pneumothorax, cardiac tamponade, and lead dislodgment was low, and there was no statistical difference among the groups. It is worth mentioning that in the last 36 cases of this series, vascular access was obtained through ultrasound-guided axillary vein puncture.

The present study has several limitations, including its observational, non-randomized retrospective nature and lack of a link to any other clinical trial. Because it is a retrospective study with a period of observation of almost 10 years, we did not have exact access to the pharmacological therapy used, which could have influenced the results.

The echocardiographic evaluation was not performed by the same observer, which might have influenced the outcome and consequently the clinical response to CRT. Finally, because the sample is relatively small and from a single service, the findings should be cautiously interpreted. Larger randomized studies are needed to confirm our data.

In conclusion, our study demonstrated that old and very old patients should not be excluded from the indication of CRT owing to their advanced age. The clinical outcomes, cardiovascular mortality, and acute complications are similar among young and elderly patients. Therefore, CRT is an effective and safe procedure in this age group.

## Supporting information

**S1 Fig. Cardiovascular survival in the 3 groups.**
(PDF)

**S2 Fig. Total survival in the 3 groups.**
(PDF)

## Author Contributions

**Conceptualization:** Luiz Eduardo Montenegro Camanho, Eduardo Benchimol Saad, Charles Slater, Luiz Antonio Oliveira Inacio Junior, Gustavo Vignoli, Lucas Carvalho Dias, Ricardo Mourilhe-Rocha.

**Data curation:** Luiz Eduardo Montenegro Camanho, Eduardo Benchimol Saad, Luiz Antonio Oliveira Inacio Junior, Gustavo Vignoli, Lucas Carvalho Dias, Pedro Pimenta de Mello Spineti, Ricardo Mourilhe-Rocha.

**Formal analysis:** Luiz Eduardo Montenegro Camanho, Eduardo Benchimol Saad, Charles Slater, Luiz Antonio Oliveira Inacio Junior, Gustavo Vignoli, Lucas Carvalho Dias, Ricardo Mourilhe-Rocha.

**Investigation:** Luiz Eduardo Montenegro Camanho, Eduardo Benchimol Saad, Charles Slater, Luiz Antonio Oliveira Inacio Junior, Gustavo Vignoli, Lucas Carvalho Dias, Pedro Pimenta de Mello Spineti, Ricardo Mourilhe-Rocha.

**Methodology:** Luiz Eduardo Montenegro Camanho, Eduardo Benchimol Saad, Charles Slater, Luiz Antonio Oliveira Inacio Junior, Gustavo Vignoli, Lucas Carvalho Dias, Pedro Pimenta de Mello Spineti, Ricardo Mourilhe-Rocha.

**Project administration:** Luiz Eduardo Montenegro Camanho, Eduardo Benchimol Saad, Charles Slater.

**Resources:** Luiz Eduardo Montenegro Camanho, Luiz Antonio Oliveira Inacio Junior, Lucas Carvalho Dias, Ricardo Mourilhe-Rocha.

**Software:** Luiz Eduardo Montenegro Camanho, Lucas Carvalho Dias, Pedro Pimenta de Mello Spineti.

**Supervision:** Luiz Eduardo Montenegro Camanho, Eduardo Benchimol Saad, Ricardo Mourilhe-Rocha.

**Validation:** Luiz Eduardo Montenegro Camanho, Eduardo Benchimol Saad, Luiz Antonio Oliveira Inacio Junior, Gustavo Vignoli, Pedro Pimenta de Mello Spineti, Ricardo Mourilhe-Rocha.

**Visualization:** Luiz Eduardo Montenegro Camanho, Eduardo Benchimol Saad, Charles Slater, Ricardo Mourilhe-Rocha.

**Writing – original draft:** Luiz Eduardo Montenegro Camanho, Eduardo Benchimol Saad, Lucas Carvalho Dias, Pedro Pimenta de Mello Spineti, Ricardo Mourilhe-Rocha.

**Writing – review & editing:** Luiz Eduardo Montenegro Camanho, Pedro Pimenta de Mello Spineti, Ricardo Mourilhe-Rocha.

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
