## [Decision Letter · Decision Letter 0]

17 Sep 2019

PONE-D-19-20560

Clinical outcomes and mortality in old and very old patients undergoing cardiac resynchronization therapy

PLOS ONE

Dear Dr. Luiz Eduardo Montenegro Camanho

Thank you for submitting your manuscript to PLOS ONE. After careful consideration, we feel that it has merit but does not fully meet PLOS ONE’s publication criteria as it currently stands. Therefore, we invite you to submit a revised version of the manuscript that addresses the points raised during the review process.

ACADEMIC EDITOR: 

Manuscript underwent two different reviewers who suggested minor (only reference's list improvement) and major revsion (regarding data that must be explained better). See my additional comment for authors below. 

Authors must be follow reviewers suggestions before manuscript acceptance.

We would appreciate receiving your revised manuscript by next 15 days (10/01). To enhance the reproducibility of your results, we recommend that if applicable you deposit your laboratory protocols in protocols.io, where a protocol can be assigned its own identifier (DOI) such that it can be cited independently in the future. For instructions see: http://journals.plos.org/plosone/s/submission-guidelines#loc-laboratory-protocols

We look forward to receiving your revised manuscript.

Kind regards,

Giuseppe Coppola

Academic Editor

PLOS ONE

Journal Requirements:

4. Please include your tables as part of your main manuscript and remove the individual files. Please note that supplementary tables (should remain/ be uploaded) as separate "supporting information" files

Additional Editor Comments:

Kind authors, thank you for submitting your manuscprit To Plos One journal. The manuscript is well written and interesting. The topic focus about elderly population undergoing CRT that is not well representad in the biggest clinical trials. You have underline how the response to CRT and the cardiovascular mortality were not affected by advanced age. Thus, advanced age should not be an exclusion factor for this therapy. I agree with you considering age as not an exclusion criteria to CRT in comparison to, for example, life expectancy and/or comorbidity burden; so, probably, discussion must be improved considering this well known findings and something about the importance role of QRS Shortening after CRT as possible variable to predict CRT response. For example you can consider our previous work "Magnitude of QRS duration reduction after biventricular pacing identifies responders to cardiac resynchronization therapy" to have a look about this topic.

I have appreciate very much when you underlined the lack of the universal definition of CRT responders and how the use of ultrasound-guided axillary or cephalic vein access has been increasingly established as a safe and effective technique related to lower rates of acute complications.

Your manuscript underwent two different reviewers who suggested minor (only reference's list improvement) and major revsion (regarding data that must be explained better).

I hope you can follow reviewers' suggestions to permit the acceptance of your interesting manuscript.

My best regards.

Reviewers' comments:

Reviewer's Responses to Questions

**Comments to the Author**

1. Is the manuscript technically sound, and do the data support the conclusions?

Reviewer #1: Yes

Reviewer #2: Yes

2. Has the statistical analysis been performed appropriately and rigorously? 

Reviewer #1: Yes

Reviewer #2: Yes

3. Have the authors made all data underlying the findings in their manuscript fully available?

Reviewer #1: Yes

Reviewer #2: No

4. Is the manuscript presented in an intelligible fashion and written in standard English?

Reviewer #1: Yes

Reviewer #2: Yes

5. Review Comments to the Author

Reviewer #1: Cardiac resynchronization therapy (CRT) is a successful strategy for heart failure (HF) patients. In the epidemiological context of recent decades, the average age of the population tends to increase, especially elderly and very elderly patients. Moreover, in the near future, it is expected a increased worldwide prevalence of heart failure. In light of this scenary, there is an urgent need for study CRT in the setting of very elderly patietns.

In the study of Luiz Eduardo Montenegro Camanho et al. it has been very interesting to have analyzed the impact of advanced age on the clinical response and cardiovascular and total mortality of patients undergoing CRT. In this regard, it would be interesting to point out that other studies have also previously evaluated clinical outcomes in elderly patients subjected to CRT (see: “Clinical outcomes in cardiac resynchronization therapy-defibrillator recipients 80 years of age and older ”, Doi: 10.1093/europace/euv222) and evaluation of importance of CRT (see: “Non-responders to cardiac resynchronization therapy: Insights from multimodality imaging and electrocardiography. A brief review”, Doi: 10.1016/j.ijcard.2016.09.037).

Returning to the manuscript, I would like congratulate all the authors contributing to this good article and research work. The study population is big enough, this is an important advantage, as it gives more statistical power. However, the population will have to be implemented to find application in the guidelines. Finally, the results were very significant and further encouraged the use of CRT in elderly patients. Conclusions of study are presented in an appropriate fashion and are supported by interesting data. The article is presented in an intelligible fashion.

In conclusion, given the overall work, I accept manuscript with minor revision concerning a small implementation of bibliography, reporting the aforementioned works and other.

Reviewer #2: There are 3 important data to explain:

- The difference in ischemic and non ischemic population in the third Group

- the difference in mortality between Crtp and Crtd

- the difference in response between population with LBBB and other wide Qrs

6. PLOS authors have the option to publish the peer review history of their article (what does this mean?). If published, this will include your full peer review and any attached files.

Reviewer #1: No

Reviewer #2: No

---

## [Author Response · Author response to Decision Letter 0]

29 Oct 2019

Dear Academic Editor and Reviewers:

Below is the answer to the comments of the reviewers:

1. Reviewer # 1:

“In conclusion, given the overall work, I accept manuscript with minor revision concerning a small implementation of bibliography, reporting the aforementioned works and other”.

The implementation of bibliography was made and the articles were inserted in the Discussion (number 25 and 27) – pages 14,15 and 20,21.

Although comparative studies do not demonstrate a direct correlation between clinical response and echocardiographic response, when a reverse remodeling criterion is associated with CRT assessment, the non-response rate was described to range from 35% to 40% [24,25]. Our results reflect this situation and the need for a more definitive classification of the problem, as when using only the FC, the response rate to CRT in the 3 groups was 82.8%, 94.9%, and 91%, respectively. When an echocardiographic reverse remodeling criterion was included, the response rate was 52.3%, 56.9%, and 68.5%, respectively.

Although there are >3500 scientific publications on ICD/CRT device implantation in patients aged >65 years, there are around 20 publications about the safety of the procedure in patients aged >75 years [26,27].

25. Caritá P, Corrado E, Pontone G, Curnis A, Bontempi L, Novo G et al. Non-responders to cardiac resynchronization therapy: Insights from multimodality imaging and electrocadiography. A brief review. International Journal of Cardiology, 225,402-407.

27. Adelstein EC, Liu J, Jain S, Schwartzman D, Althouse AD, Wang NC, et al. Clinical Outcomes in cardiac resynchronization therapy-defibrilaltor recipients 80 years of age and older. Europace, 18(3), 420-427.

2. Reviewer # 2:

“There are 3 important data to explain”

- “The difference in ischemic and non ischemic population in the third Group”

This paragraph below was inserted in Results/ Primary endpoint: cardiac resynchronization therapy response and mortality (page 11).

In this group (>75 years), ischemic etiology, defined as previous myocardial infarction, PTCA, myocardial revascularization or severe coronary obstructive disease, was observed in 53.9%. No clinical differences and outcomes were observed between ischemic and non-ischemic very old patients. 

- “The difference in mortality between Crtp and Crtd”

This paragraph below was inserted in Results/ Primary endpoint: cardiac resynchronization therapy response and mortality (page 11).

CRT-D was the prosthesis used in 75% of the patients approximately. When survival curves were analyzed according to the type of prosthesis, it was observed that CRT-p or CRT-d were not predictors of cardiovascular or total mortality, with similar survival rates.

- “The difference in response between population with LBBB and other wide QRS”

This paragraph below was inserted in Results/ Primary endpoint: cardiac resynchronization therapy response and mortality (page 11).

The LBBB occurred in 93%, 94% and 98%, respectively, in the three groups. In 12/ 249 pt (4.8%) was observed right bundle branch block (RBBB), 6 in group I, 2 in group II and 4 in group III. The QRS width average was 167 ms. Except one patient in group I, all RBBB pt were responders to CRT.

---

## [Editor Report · Decision Letter 1]

8 Nov 2019

Clinical outcomes and mortality in old and very old patients undergoing cardiac resynchronization therapy

PONE-D-19-20560R1

Dear Dr. Luiz Eduardo Montenegro Camanho

We are pleased to inform you that your manuscript has been judged scientifically suitable for publication and will be formally accepted for publication once it complies with all outstanding technical requirements.

With kind regards,

Giuseppe Coppola

Academic Editor

PLOS ONE

Additional Editor Comments (optional):

Kind Author, after careful new evaluation I can say that your manuscript underwent full revision point by point according with reviewer's suggestions.

---

## [Editor Report · Acceptance letter]

21 Nov 2019

PONE-D-19-20560R1 

Clinical outcomes and mortality in old and very old patients undergoing cardiac resynchronization therapy 

Dear Dr. Montenegro Camanho:

I am pleased to inform you that your manuscript has been deemed suitable for publication in PLOS ONE. Congratulations! Your manuscript is now with our production department. 

With kind regards,

on behalf of

Dr. Giuseppe Coppola 

Academic Editor

PLOS ONE